# Prehistoric Recycling Explained in a Playful Way: The *Pfahlbauten Wimmelbild*—An Interactive Digital Mediation Tool Designed by Young People

**Helena Seidl da Fonseca [1], Fiona Leipold [1],\*, and Karina Grömer [2],\***

1　Kuratorium Pfahlbauten, 1010 Vienna, Austria; seidl@pfahlbauten.at
2　Naturhistorisches Museum, 1010 Vienna, Austria
\*　Correspondence: leipold@pfahlbauten.at (F.L.); karina.groemer@nhm-wien.ac.at (K.G.)

**Abstract:** With the "Talents Internship program" established by the Austrian Research Promotion Agency (FFG, Österreichische Forschungsförderungsgesellschaft), 14- to 17-year-old students from various school types visited the Natural History Museum and the Kuratorium Pfahlbauten in summer 2022 to gain practical experience in research. The internship focused on a sustainability approach, discussing recycling methods, the sustainable use of resources and the circular economy in prehistory. The UNESCO World Heritage "Prehistoric Pile Dwellings around the Alps" was used as a research area for the project. The project also aimed to make the content developed by the students available to the public as a digital media tool. The pupils brought an illustration of a prehistoric lake shore settlement to life and created an interactive image available at the website of Kuratorium Pfahlbauten. Various scenes of the illustration have been augmented with animations created by the students of HTL Spengergasse in Vienna. Students from federal secondary schools from Vienna (Stubenbastei) and Upper Austria (Traun) researched the information about the objects and wrote texts that, as a description of the animated videos, introduce the users to the prehistoric artifact and explain the recycling process behind it. The students worked independently using the scientific literature, 140-year-old inventory books and 6000-year-old objects from the collection of the Natural History Museum Vienna. The activities and the supporting program within the internship were recorded by the students in blog posts, available at the Pfahlbauten-Blog. The co-creative approach of the FFG Talent Internship made it possible to introduce a group of school students to the process of scientific work and the communication of results. It was honored with the Creative App Award at CHNT 2023.

**Keywords:** digital mediation tool; prehistoric pile dwellings; UNESCO World Heritage; prehistoric recycling methods; science meets school; students internship; youth development

## 1. Making Cultural Heritage Accessible to Young People

Archaeological research needs a strong commitment to public outreach and education to make research results and the understanding of science accessible not only to a scientific audience but also to the general public. Since archaeological research is mostly financed by taxpayers' money, one could argue that it is even an obligation to publish and communicate the results to the public. This involves a wide audience with various focused interests, from children to adults, non-experts, scientists and academic groups. The Natural History Museum Vienna is one of the largest non-university research centers in Austria. With a variety of exhibits, different research departments at the museum enable interdisciplinary and diverse research. Research in the 21st century, as well as its mediation, needs to be integrated into the challenges of our time such as the use of resources, access to cultural heritage, the perpetuation of techniques and skills belonging to intangible heritage, and people's identity, mobility or migration [1].

The knowledge transfer activities organized by the Natural History Museum, and especially the Department of Prehistory, are very diverse. The academic exchange of the

museum's research results and expertise includes publications, conferences and teaching at various universities, in addition to the support of guest researchers and students who are working with materials from the Prehistoric Department as part of their academic work. The research results are also disseminated to the general public through exhibitions [2]. The Natural History Museum Vienna set up a space to share and co-create information called "Deck 50" (see Figure 1) where people are invited to become involved in scientific research [3], which includes a Citizen Science approach in those activities as well [4]. Also, lectures are important for science communication, as are press works such as radio interviews, television interviews, blogs [5], interviews in newspapers, podcasts, YouTube videos #nhmfromhome [6], Instagram appearances, etc.

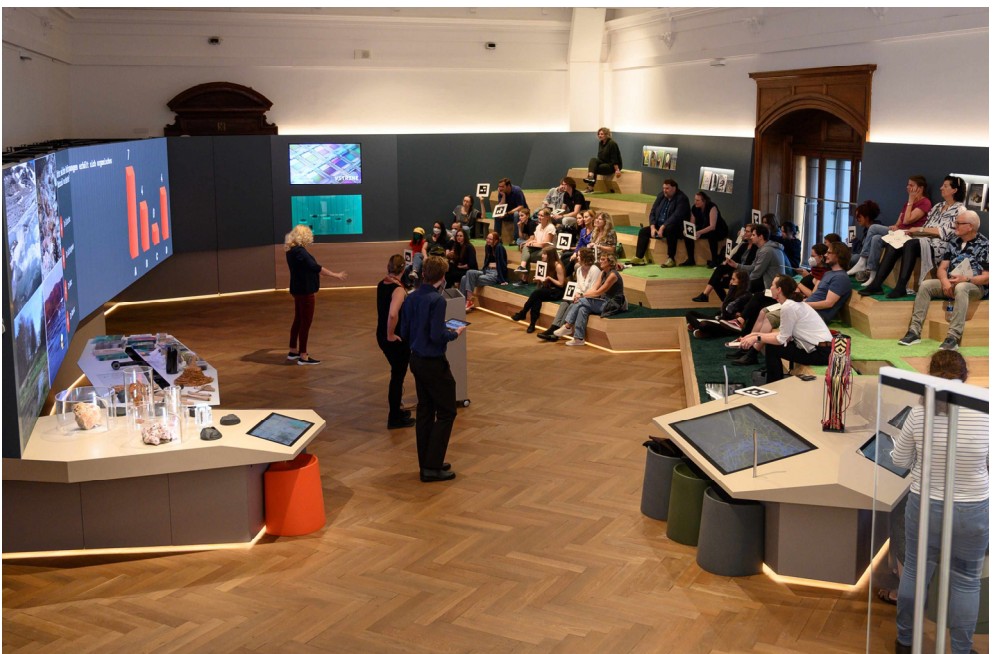

**Figure 1.** "Deck 50"—a new dissemination room at the Natural History Museum Vienna ©. Natural History Museum Vienna.

Research content is also conveyed in events, e.g., "*Archäologie am Berg*" (archaeology on the mountain) at the Salzwelten in Hallstatt (see Figure 2), the Long Nights of the Museums, historical fashion shows, etc., as well as in the context of Young Science Ambassador activities [7] and internships for interested schoolchildren. In the various activities, comprehensible communication of research content, inclusion and participatory elements are important for us, as well as references to topics of contemporary significance. Aside from this, the Prehistoric Department also focuses on digital dissemination on various levels. The internet has added impetus here, as information on archaeological artifacts is more easily available and enthusiasts for ancient objects and past lives can be reached across the globe.

This also involves collaborative activities such as the *Pfahlbauten Wimmelbild* together with the Kuratorium Pfahlbauten. The Kuratorium Pfahlbauten is a state-funded association responsible for the management of the Austrian part of the UNESCO World Heritage "Prehistoric Pile Dwellings around the Alps". UNESCO World Heritage Management has a clear mission and the intention to provide access to the World Heritage Site of Prehistoric Pile Dwellings and create opportunities for participation [8,9] (p. 1069). This is often a major challenge. Although the prehistoric village ruins on Lake Attersee, Lake Keutschach and Lake Mondsee, which are up to 6000 years old, are invaluable from an archaeological perspective and are of exceptional and universal value, this value is difficult to communicate to the broader public. Not only are the human remains, consisting of layers of settlement waste and construction timber, often visually unspectacular to the layman,

but due to their location under water (which makes it possible to preserve the finds in the first place), they are simply invisible (see Figures 3 and 4). All Austrian sites are located in lakes and can only be viewed underwater by diving or with special cameras. This makes access and thus understanding the value of these sites extremely difficult [10,11].

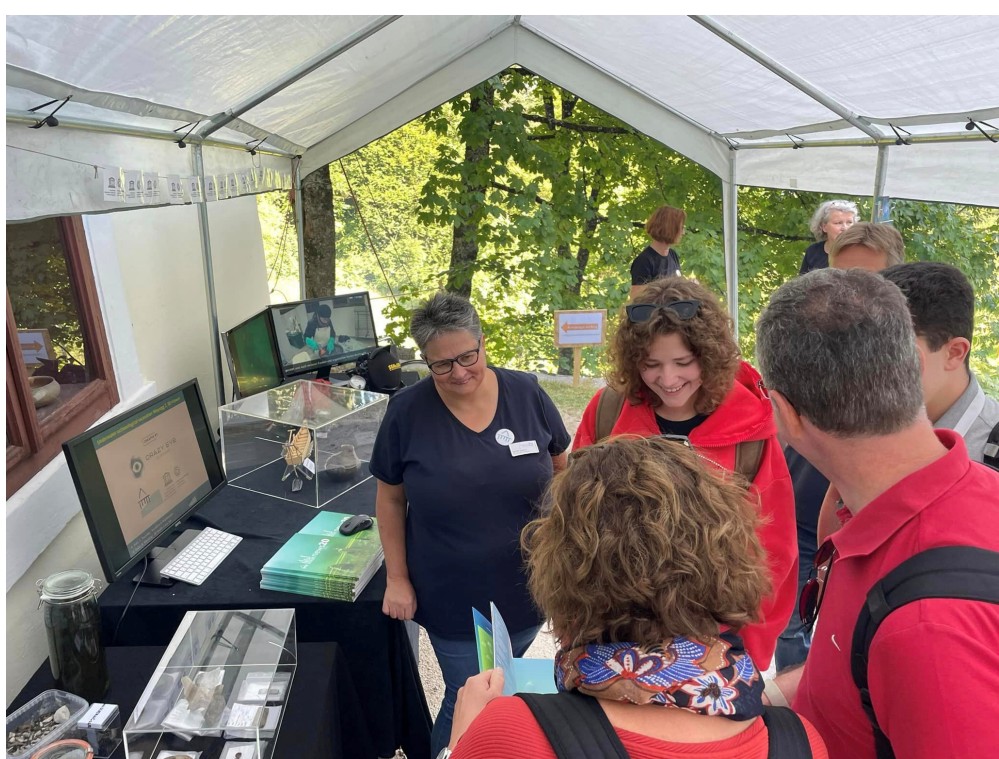

**Figure 2.** Visitors come into contact with scientists at the event "*Archäologie am Berg*" in Hallstatt ©. Kuratorium Pfahlbauten.

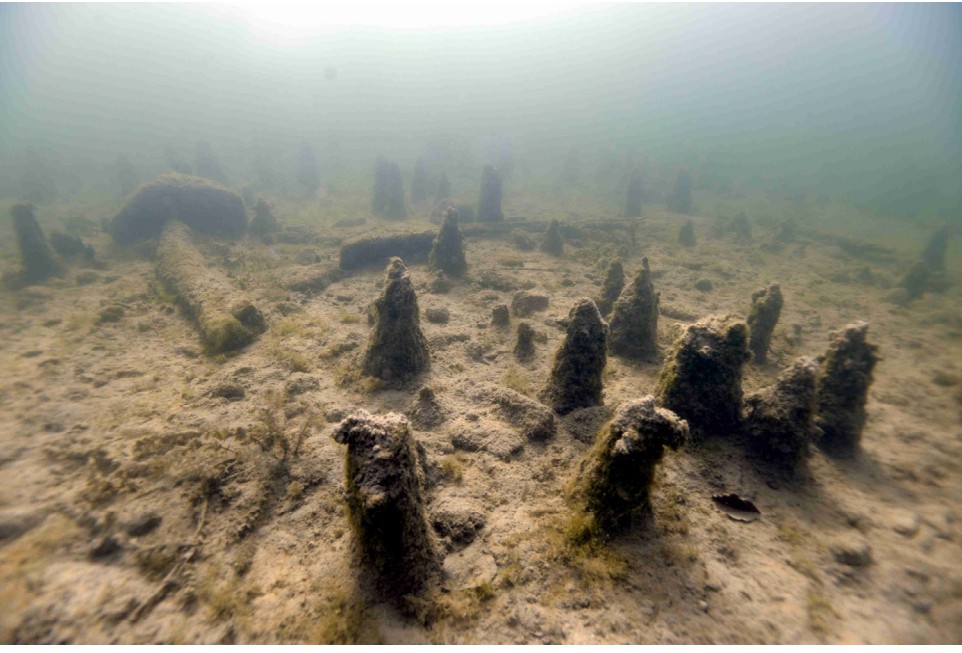

**Figure 3.** UNESCO World Heritage Site *See am Mondsee* at Lake Mondsee in Upper Austria. © Kuratorium Pfahlbauten.

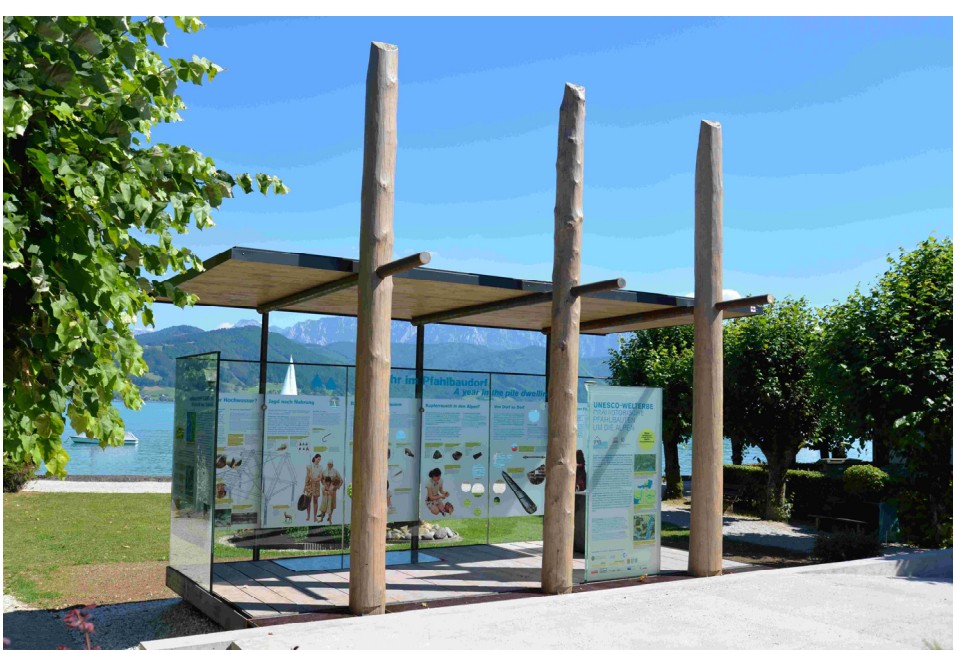

**Figure 4.** Information pavilion of the UNESCO World Heritage "Prehistoric Pile dwellings around the Alps" at Lake Attersee, Upper Austria. © Kuratorium Pfahlbauten.

The Kuratorium Pfahlbauten has made it one of its main tasks to create low-threshold, exciting and innovative approaches to the World Heritage Site of Prehistoric Pile Dwellings. These should appeal to different sections of the population, such as varying age groups and interests. Projects such as the *Pfahlbauten Wimmelbild* provide a playful and interactive introduction to the topic of pile dwellings and, with topics such as recycling and the use of resources, tie in with the world in which children, young people and adults live today (see Figure 5).

The co-creative approach of the FFG Talent Internship also makes it possible to introduce a group of school students to the process of scientific work and the communication of results. They get a feeling for the problems of mediation and public communication. Conversely, scientific institutions such as the Natural History Museum Vienna and the Kuratorium Pfahlbauten can gain an outside perspective from the young generation and expand their educational programs with their input.

Within the framework of the project "Prehistoric recycling: Sustainable resources and recycling in prehistoric times", pupils were asked to use archaeological artifacts from the UNESCO World Heritage Site "Prehistoric Pile Dwellings around the Alps" to study how raw materials and waste were handled in prehistory. Addressing the challenges of our time with our deep history perspective can be exemplified with the topic of resource management. Various artifact groups can be rediscovered as remnants of a widespread recycling economy, e.g., textiles, stone tools, ceramics, food leftovers, etc. [12], and exemplars of the careful and efficient use of resources. How people handled objects sensibly and sensitivy a long time ago, with reuse and recycling, can be a model for today's discourse. Regarding resource usage, it is possible for us to learn from the past.

*Seven Teenagers, Lots of Ideas!*

Within the FFG Talent Internship, the focus of the research work was on finding evidence of recycling and reuse in a time 6000 years ago to gain a glimpse into the sustainable use habits of prehistoric populations. Based on selected example objects from a wide spectrum of artifacts from prehistoric pile dwellings around the Alps, prehistoric behavior and creativity in dealing with raw materials and recycling were to be investigated. The students' work should result in a product that can be used for science communication—the *Pfahlbauten Wimmelbild*.

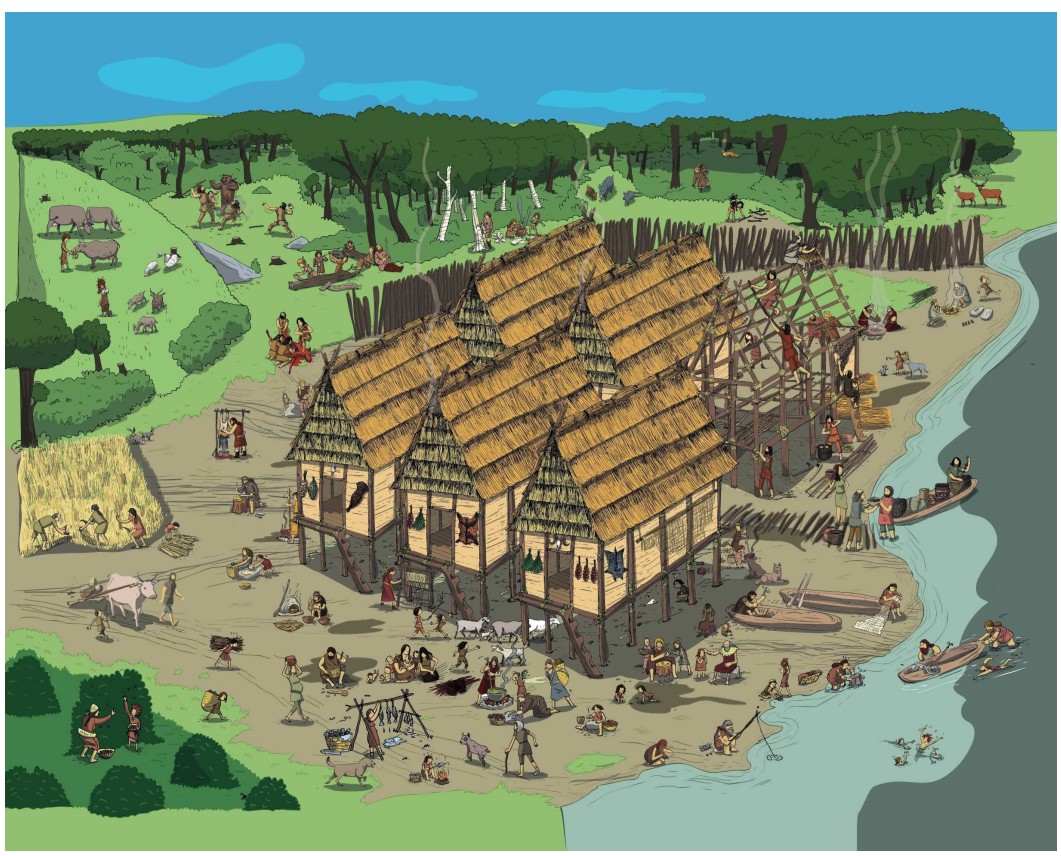

**Figure 5.** The graphic basis—a playful illustration about Neolithic life in a pile-dwelling village. © Kuratorium Pfahlbauten. Illustrator: Leopold Maurer.

From 11 July to 5 August 2022, seven students, aged 14 to 17, visited the Prehistoric Department of the Natural History Museum Vienna for this task: F. Doubrava, A. Ferrufino Uria, L. Jona and S. Knittl from HTL Spengergasse (Vienna), M. Lonsing from BRG Traun (Upper Austria), and Z. Mund and M. V. Schwarcz from GRG Stubenbastei (Vienna). The interns were supervised by Karina Grömer (Natural History Museum Vienna), and Helena Seidl da Fonseca and Fiona Leipold (Kuratorium Pfahlbauten).

In Austria, a student internship in a company or institution is part of public school education. The internships are subject to certain regulations. For example, the pupils may only be commissioned with light activities that are suitable for them. However, those who offer an internship should ensure that the students become familiar with every day working life in their profession. Independent work according to prior guidance is desirable. After the internship, the participants receive an internship certificate and often have to summarize their activities in a report for the school. Every year, the Natural History Museum Vienna supervises various student internships. Most students come from Vienna, but especially in the summer months, students from the surrounding provinces also complete an internship at the capital's Federal Museum. As part of the FFG Talent Internship, additional requirements must be met to receive funding for four weeks. To this end, projects on the annual focus topics must be submitted to the funding scheme in advance. The students who signed up for the FFG Talent Internship in the Prehistory Department were mainly interested in archaeology, history and genealogy. For example, the student from Upper Austria had learned to read an old form of a German handwriting script type ("Kurrent" or "German cursive") in the course of exploring his passion and was ideally suited to work on the museum's inventory books. In the up to 140-year-old museum's inventory books, there are entries for each object from the prehistoric collection. In the collection, many objects from the sites of prehistoric pile dwellings originate from

investigations of the 19th century. Therefore, it was clear from the beginning that entries on selected objects would have to be researched in the course of the project.

The students of the HTL Spengergasse have a focus on media design in their education. They learn special skills in the areas of 2D and 3D animation, motion design, video, visual effects, concept art, sound design and dramaturgy. Therefore, they had an interest in presenting the objects and the scientific findings about them in a modern and playful way.

Even before the project was submitted, meetings between the scientists and the students were necessary to explore the scope and opportunities for the students within the internship. The preliminary meetings were decisive for the planning of the project with regard to the assessment of the need for care and other necessary help from outside experts. It also enabled a discussion on the planned focus topic and to jointly develop a creative product to convey the scientific content.

The idea arose to create an interactive image of a prehistoric pile-dwelling settlement designed for children as a web application (see Figure 6). This mediation tool offers the possibility of implementation on websites and use in showrooms or at events on portable devices. The web app is intended to encourage visitors to interact with the "picture of past life" and to convey the content on prehistoric recycling processes in a playful way. The HTL students, who have special computer skills as professional animation design programmers, have brought various scenes from the illustration to life with small preview animations. These are designed to encourage viewers to click on the moving image to open a new section with a longer animation version and to see the objects presented in more detail, with evidence and an explanation of prehistoric reuse.

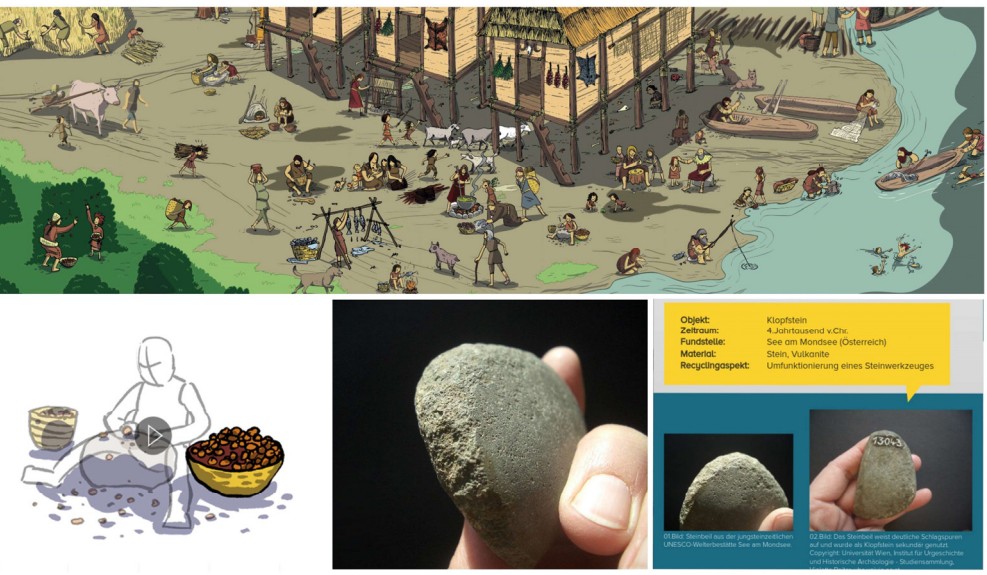

**Figure 6.** The *Pfahlbauten Wimmelbild*—https://www.pfahlbauten.at/wimmelbild (accessed on 22 August 2024) illustration © Kuratorium Pfahlbauten, illustrator: Leopold Maurer, object images © Study Collection of the Institute of Prehistory and Early History of the University of Vienna, photographer: Violetta Reiter.

It quickly became clear that the students were divided into two working groups: the one working on the technical implementation (HTL students) and the one working on the necessary content (secondary school students)—the description of recycling processes of eight copper-age artifacts and the selection of images. The preliminary discussions were of great importance for the successful execution of the project. They enabled the supervisors to assess the students' abilities and the effort required for performing all tasks, and to plan the necessary workshops.

## 2. Working Methods and Technical Implementation

The challenge was to work out the text explaining the recycling process in a concise and understandable way. Also, the design of the short animations from the given illustration was challenging. The animations had to show people performing an activity on which the suitable text about the recycling process of the object can be based.

In order to quickly and successfully implement the planned mediation tool, the colleagues of Kuratorium Pfahlbauten undertook the preliminary research on prehistoric objects from pile-dwelling settlements with traces of recycling processes. Objects were chosen that show patterns of reuse during prehistoric times, like a stone axe blade from the site See at Lake Mondsee (Austria). The stone tool was crafted as an axe, but over time, the blade became blunt. When the tool itself became too small to be sharpened, it was converted into a tapping stone. This reuse is clearly visible on the stone object through striking marks [13]. Another famous stone tool from a pile-dwelling site in Germany, the flint dagger from Allensbach, tells a long story of repair work and shows how far objects traveled in prehistory [14]. But not only stone objects have been re-used. Ceramics from the pile-dwelling site Arbon Bleiche 3 (Switzerland) show fireclay tempering. This means that crushed clay shards were mixed into the clay mass. Pile dwellers thus reused old ceramic remains even in their time [15]. The selection of objects representing prehistoric recycling methods includes eight objects made of different materials, such as stone, ceramics, antler or textile, and demonstrates the exceptional preservation conditions at the sites in wet soil. Helena Seidl da Fonseca, underwater archaeologist and specialist in pile-dwelling settlements, assisted the students in digging up the literature and obtaining the image rights to the objects for the web app. Filtering out interesting information about the objects from scientific publications, as well as the actual writing of the texts, was carried out surprisingly independently by the students (see Figures 7 and 8). Some of them worked with 140-year-old inventory books and sought help from other departments of the museum in transcribing the description of the artefacts written in "Kurrent" script. They developed such an interest and desire to write that they published their work processes and experiences during the internship in humorous blog posts on the *pfahlbauten.at/blog* (accessed on 22 August 2024).

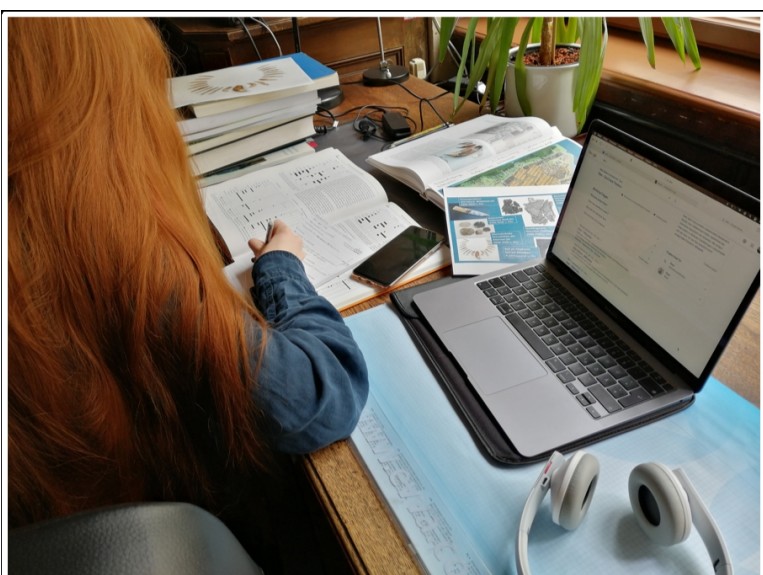

**Figure 7.** Student digging up interesting information about prehistoric objects from scientific publications. © Kuratorium Pfahlbauten.

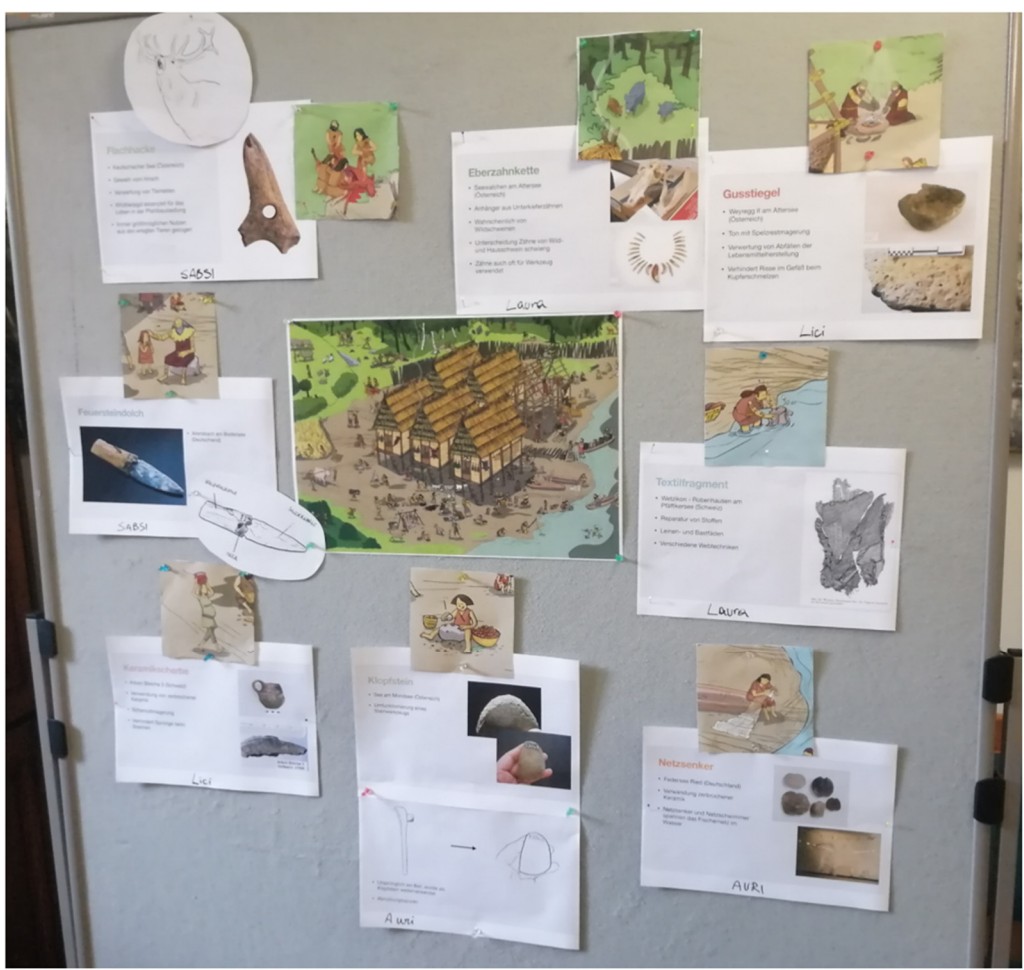

**Figure 8.** The mind board—putting all of the puzzle pieces together. © Kuratorium Pfahlbauten.

The supervision of the group of students for the technical implementation had to be taken over by external experts. In the first week of the project, a workshop was held in collaboration with the Vienna-based illustrator Leopold Maurer. The original image *Pfahlbauten Wimmelbild* was created by him and lovingly worked out down to the smallest detail. The illustrations served as source material for the students to create their animations (see Figure 9). He provided his vector graphics for further processing and discussed the individual steps of their planned sprite animations with the students. The workshop was not only about the technical work steps, but also about creative brainstorming on how human movements or emotional expressions can be implemented in soundless, graphic form. At the end of the workshop, the short animations were developed independently by the students. The animations were created using the Toon Boom software (see Figure 10). The software is not able to export a vector-based format, which is why classic sprite animations were used, like those known from old game consoles like the NES. The sprites were animated using the Konva.js library in Canvas objects. The web app was implemented in cooperation with the company lowfidelity Heavy Industries. Ingo Zehenthofer and Sascha Szojak programmed the digital mediation tool and incorporated the animations and texts of the students into the web app. The final *Pfahlbauten Wimmelbild* is a very large illustration and shows many small life scenes of a prehistoric settlement. It is therefore necessary to zoom in on the image at any point to be able to view scenes closely. The user experience was based on online map functionality such as Google Maps.

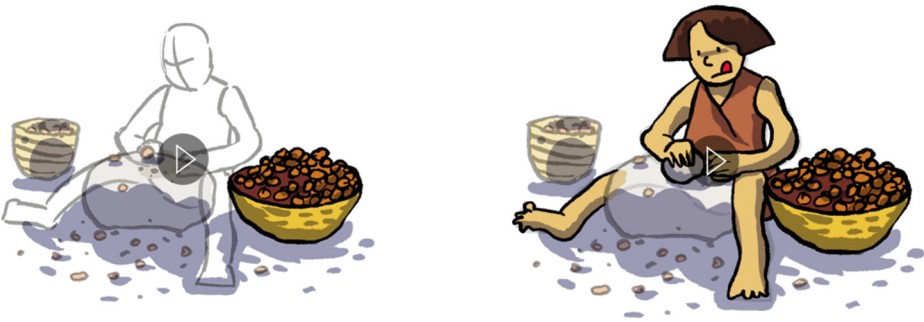

**Figure 9.** Scenes from the illustration were redrawn by the students and served as the basis for the animations. © Kuratorium Pfahlbauten.

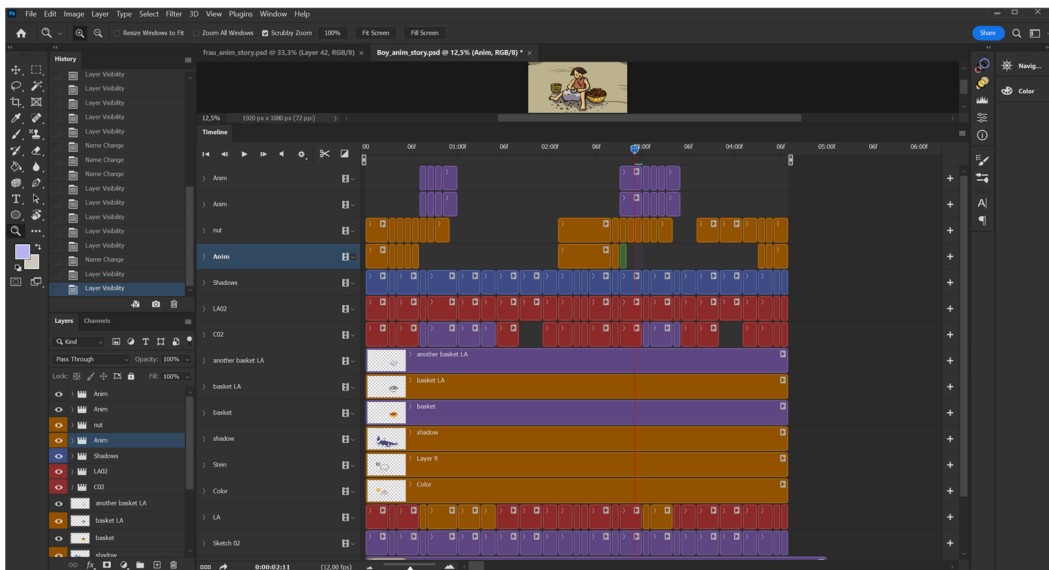

**Figure 10.** The students created the short animations independently using the software Toon Boom. © Kuratorium Pfahlbauten.

## 2.1. Educational Approach and Didactic Implementation

Within the FFG talent internship, the students are supposed to obtain practical experience in research and in the day-to-day activities of a scientist. The literature research and the examination of scientific analyses on the genesis of a prehistoric object are essential parts of an archaeologist's work. Furthermore, the communication and visualization of the results are important parts of a scientist's job, especially for those who work in museums or want to create access to an invisible UNESCO World Heritage. During their internship, the students were able to gain experience in the development and implementation of a mediation tool of scientific content. They learned to let their work package go through correction loops and to submit it revised at the end by a certain deadline.

The main task of the internship supervisors was to guide the individual work steps and to ensure coordination between the different working groups during the development process of the product. For example, weekly project meetings were scheduled in which the students presented their work progress, their problems and possible solutions. After four weeks, their work and the final product were presented to a wide audience; parents, teachers and museum staff were invited for the first presentation and release of the web application in the Natural History Museum's science communication room "Deck 50" (Figures 11 and 12). Due to the time limit of four weeks, an evaluation and improvement loop of the web application by the students could not be implemented. However, the installation of the interactive *Pfahlbauten Wimmelbild* platform is planned for the new website of the Kuratorium Pfahlbauten (planned release 2025). By recording the online

views and which content is clicked on, an evaluation of the mediation tool is possible and serves as a basis for improvements to the tool and for better planning of new products.

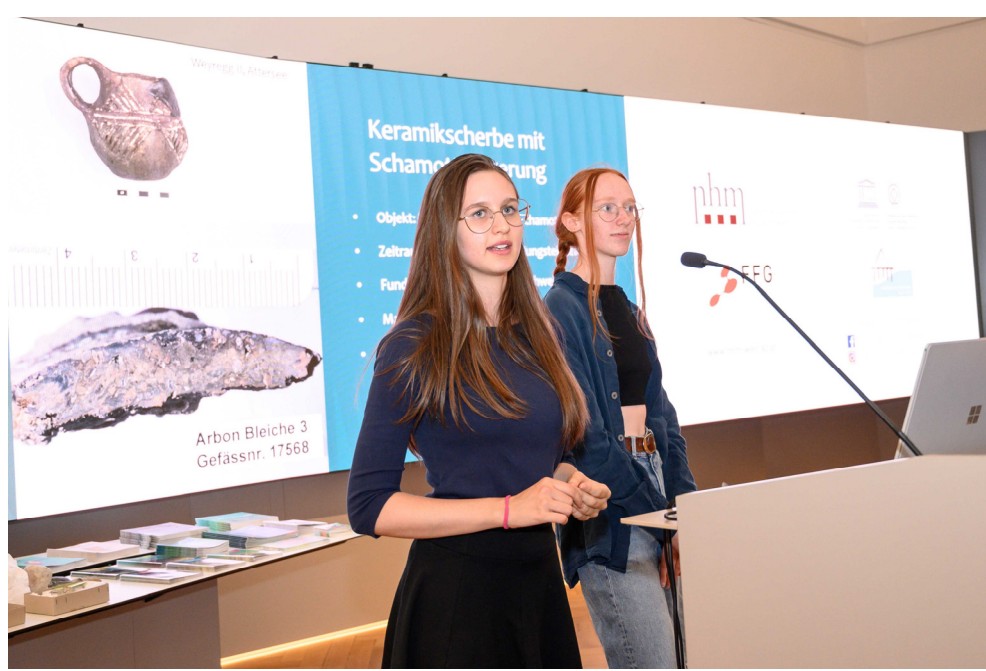

**Figure 11.** Students presenting the final product and their working steps. © Natural History Museum Vienna.

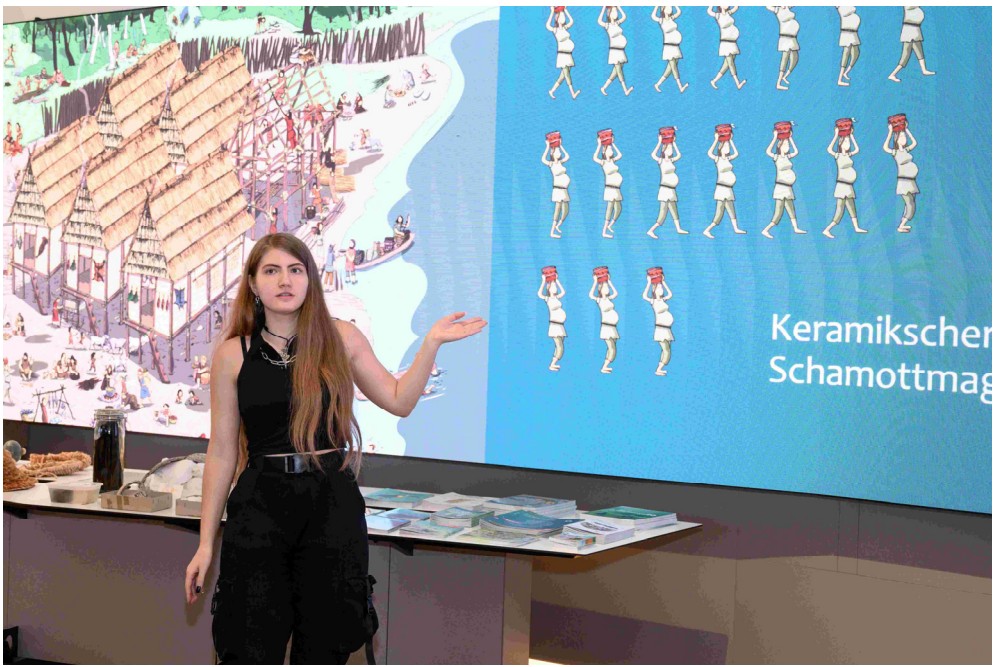

**Figure 12.** The publication of the web application was presented in the science communication room "Deck 50" of the Natural History Museum. © Natural History Museum Vienna.

### 2.2. Through the Innermost Part of the Museum

In order to give the students not only the opportunity to work on a project to convey science but also to gain a deeper understanding of the structures and working methods of a research and exhibition institution, the internship was accompanied by a comprehensive supporting program. The museum has over 270 years of history, emerging from the

imperial collections of the 18th century, when it was dedicated to the "realm of nature and its exploration". The central tasks of the museum were always the collection, preservation, research, presentation and communication of its objects and research outcomes. The challenges of the tasks of preserving, expanding and making accessible the extensive biological, geoscientific, anthropological and archaeological collections with more than 30 million objects are to be brought closer to the students, as well as current scientific methods and their constant development. Besides its specialist departments with extensive collections and scientific staff, the Natural History Museum has the necessary infrastructure for the entire operation of this huge scientific institution, such as carpentry, taxidermy, etc. When it was founded in 1876, the museum was already designed as a networked research and education institution.

To give the students an understanding of these processes and the long history, they spent an average of four hours each week visiting different specialist departments or going on background tours in the museum. They were able to talk to colleagues from the anthropology department, identify animal bones in the archaeozoology department, look over the shoulders of the taxidermists in their workshop, learn about 19th century research and documentation in the archives and also test the new and innovative communication possibilities of "Deck50". Experienced building technicians and scientific staff took the students on a backstage tour and thus into areas that hardly any other visitors get to see. From the cellars and old ventilation shafts to the depot, which extends five floors down and contains the majority of the collections, and to the roof with a view over Vienna's Ringstrasse, everything was there.

The students were able to process their experiences and impressions directly in detailed blog posts and prepare them for a wider audience. In total, the reports on their work on the *Pfahlbauten Wimmelbild* project resulted in a series of 15 blog posts available at https://www.pfahlbauten.at/blog (accessed on 22 August 2024). These were promoted via the Facebook account of the Kuratorium Pfahlbauten, among others, where further short reports and insights into the students' work could also be seen. The students were given free rein on Instagram. With minimal input and assistance, they were allowed to design, create and implement an Instagram campaign to accompany the internship. In it, they introduced themselves and their fellow students as well as the object they worked on for the *Pfahlbauten Wimmelbild*.

## 3. Output and Positive Side Effects

The *Pfahlbauten Wimmelbild* is useful for different purposes. On the one hand, it is available as a web application (web app) via the website of the Kuratorium Pfahlbauten, and on the other hand, it serves as an interactive installation for exhibitions. The hidden object is operated via a touch display or tablet and transmitted to large screens or projected onto screens. It can be easily implemented in small exhibitions and can be operated by the visitor and without any additional explanation. The animations created by the students are very versatile, so if it should be necessary or required to change the topic at any time, only the content of the subpages has to be changed. This means that in order to adapt the tool to specific exhibitions or events, the web application content can be easily filled with new topics other than prehistoric recycling, without changing the visual aspect. It is also possible to add animations into the picture, if necessary. In other words, the web application is adaptable and expandable. This mediation tool would not have been implemented as a commissioned work by an illustrator or a company for the production of such products. The cost of working time to create the animations would not have been in the budget of the museum or the Kuratorium Pfahlbauten. Therefore, the internship was not only an enriching experience for the students, but also served as a high-quality support in product development implementation of a mediation tool for science communication. The cooperation with the students shows that internships can be fruitful for both sides, if there is an intensive exchange between both parties in the planning and design phase.

The *Pfahlbauten Wimmelbild* can be used as an eyecatcher and self-operated mediation tool at events involving the public (e.g., Long Nights of the Museums, European Researchers Night, *Archäologie am Berg*, World Heritage Day, etc.). The use of the tool was tested during the event Long Nights of the Museums on 1 October 2022. Visitors were able to try out the web application on tablets at an information table of the prehistoric department. In particular, children and young adults were attracted by the *Pfahlbauten Wimmelbild* and its presentation on new media. They were more than willing to try out the web application and discover the content on their own. Since then, the mediation tool has been used regularly at events, such as the 3rd Austrian World Heritage Day on 18 April 2023. The innovative approach of this project and the work with school kids was honored by the Creative App award in 2023 at the CHNT conference in Vienna (CHNT-ICOMOS Austria, 2023).

The project, which involved almost exclusively female students, also attracted the attention of the Soroptimist International organization. This organization for the advancement of women enabled the students to go on a two-day excursion to the field office of the Department of Prehistory from the Natural History Museum Vienna at the prehistoric salt mine Hallstatt as part of the internship. The excursion enabled the young students to visit one of the most important archaeological sites in Austria and to come into contact with the scientists on site.

## 4. Students Internship—Is It Worth the Effort and Time?

The benefits of the involvement of Citizen Scientists in archaeological research and the necessity of divulging information and of communication with different sectors of the public are well known in the archaeological field. Projects such as "united by crisis" show us how interested citizens can be included in the research of their own region and how they can contribute to expanding the site database through their knowledge and commitment [16]. In Austria, various initiatives, like Culture Connected, Sparkling Science, the FFG Talent Internship and others, also try to promote the integration of the population into scientific work at a young age. Working with children and schools can be challenging. It requires good communication and logistics, a feel for people, and a certain amount of willpower to integrate the sometimes conflicting daily routines into each other's everyday lives, especially for projects that were realized during the school semester, such as the Sparkling Science projects *Holz für Salz* [17] or *Doing Welterbe—Welterbe begreifen* [18].

There is no question that the planning, supervision and support of the students during an internship is time-consuming. Working with young and unexperienced people requires a certain degree of involvement. The project managers already had this experience during the FFG Talent Internship *Visualisierung von Kleidung in der Urgeschichte* carried out in 2020 [19]. The project results of the internship in 2022 did not bring a significant improvement in scientific knowledge, but the result is a versatile mediation tool, whose content can be easily redesigned for new purposes. It is an actual product that can be used to convey content related to the prehistoric pile dwellings on various occasions. In addition to the actual output and its public utilization, PR support for the process is a key factor in raising public awareness. Science and the organizations involved show that they are open, accessible and supportive toward the young generation and, at the same time, involve them in the public relations process.

In the implementation of the project, however, the focus was less on the end result and more on the communication of scientific work to the participants themselves. In the individual steps, it was possible to successfully convey the logic of the scientific subject to the participants. While dealing with the objects and the literature, they gained knowledge about scientific methods of archaeology, which they needed in order to write educational texts. The independent work and the encouragement to develop one's own creativity acted as a strong motivator for the students to work through even the most unexciting scientific texts. The internship gave the students actual insights into scientific work and enabled the students to implement a usable mediation product that would not have been created in this form without their work. A well-thought-out and didactically prepared student internship

is one of the best ways to attract and interest young people in science, research, history and our heritage.

**Author Contributions:** Conceptualization, H.S.d.F.—Kuratorium Pfahlbauten, F.L.—Kuratorium Pfahlbauten and K.G.—Naturhistorisches Museum Wien; methodology, H.S.d.F.—Kuratorium Pfahlbauten and Ingo Zehenthofer—lowfidelity HEAVY INDUSTRIES; software, Toon Boom and Konva.js Libraries; validation, FFG—Die Österreichische Forschungsförderungsgesellschaft; formal analysis, H.S.d.F. and F.L.—Kuratorium Pfahlbauten; investigation, H.S.d.F.—Kuratorium Pfahlbauten; resources, FFG Talente Praktikum; data curation, H.S.d.F.—Kuratorium Pfahlbauten; writing—original draft preparation, H.S.d.F.—Kuratorium Pfahlbauten, F.L.—Kuratorium Pfahlbauten and K.G.—Naturhistorisches Museum Wien; writing—review and editing, Cyril Dworsky—Kuratorium Pfahlbauten; visualization, Ingo Zehenthofer—lowfidelity HEAVY INDUSTRIES, Sascha Szojak—lowfidelity HEAVY INDUSTRIES and Leopold Maurer—Illustrator; supervision, K.G.—Naturhistorisches Museum Wien and Cyril Dworsky—Kuratorium Pfahlbauten; project administration, K.G.—Naturhistorisches Museum Wien; funding acquisition, K.G.—Naturhistorisches Museum Wien. All authors have read and agreed to the published version of the manuscript.

**Funding:** This project was funded by FFG. https://www.ffg.at. The FFG is the central national funding organisation and strengthens Austria's innovative strength. Funding number 44417304 in the FFG's call for proposals "Talent Internships" in 2022, category "Circular Economy", project title: *Ur-Recycling—Nachhaltige Ressourcen und Recycling schon in der Urgeschichte.*

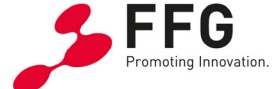

**Data Availability Statement:** The photos of the archaeological objects are subject to the image rights of the respective collection to which they belong and cannot be downloaded. The animations created by the students originate from the illustration of the *Pfahlbauten Wimmelbild* and are property of the Kuratorium Pfahlbauten. However, the interactive digital mediation tool is available online at https://www.pfahlbauten.at/wimmelbild (accessed on 22 August 2024) and can be used for free.

**Acknowledgments:** A special thanks goes to the organization Soroptimist International, which supported the young female participants and enabled them to round off their internship with an excursion to the archaeological site of Hallstatt.

**Conflicts of Interest:** The authors declare no conflicts of interest.

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
