# Peer review of "Prehistoric Recycling Explained in a Playful Way: The Pfahlbauten Wimmelbild—An Interactive Digital Mediation Tool Designed by Young People"

_heritage, doi:10.3390/heritage7090217_

Round 1
Reviewer 1 Report
Comments and Suggestions for Authors
The article is clearly written and the research is original and crucial to better understand prehistoric heritage in a playful way to pupils.
References are adequate and relevant.
Final outputs might show both positive effects and future horizon in a more detailed way.
Author Response
Comment 1: The article is clearly written and the research is original and crucial to better understand prehistoric heritage in a playful way to pupils.
Response 1: Thank you, we appreciate it.
Comment 2: References are adequate and relevant.
Response 2: Thank you, we have adjusted the bibliography after our first review from the CHNT conference proceedings. We are happy, that it is fitting now.
Comment 3: Final outputs might show both positive effects and future horizon in a more detailed way.
Response 3: See new content from 406-426.
Reviewer 2 Report
Comments and Suggestions for Authors
I found the article necessary and coherent. Above all, it is a presentation of a scientific and cultural mediation project. I can only emphasise the need for this type of work and the fact that articles are written to present it. I was particularly interested in the subject.
Overall, I considered the paper to be well-written, and I would like to make a few comments to improve the reading of the article and provide it with a better context.
1: The plan consists of 4 main parts. Two of them have sub-sections, but I don't understand why there are paragraphs in these parts that don't belong to sub-sections. I think you should rethink this aspect of your paper so as to achieve a better balance between parts and sub-parts.
2: Nevertheless, it is still easy to follow the development of your ideas and they remain very clear. One aspect of your plan that should also improve is the place you give to the museum's presentation. The method refers to the work done on 140-year-old inventories. It is only when the museum is presented in detail in the following sub-sections that we understand why this work is being carried out on these old inventories. The presentation of the museum could therefore come before and be inserted into the sections presenting the context in order to better understand the method used by the students.
3: A few sentences about the background of the students taking part in the project might help us to understand why they chose to participate and do this internship. For readers unfamiliar with your school system, it is important to specify whether the students are in private or state education. This is essential if we want to analyse the scientific mediation process that has been developed here and help other institutions to draw inspiration from your experience.
4: A parallel could also be drawn with similar projects.
5: A few details:
- lines 33-34: I fully understand the point being made, but I find it a doubly sharp argument to use this phrase, which is often employed to criticise archaeology and heritage protection institutions. The argument could be turned more towards the fact that archaeology and heritage are the property of everyone, and that the work of the archaeologist is by its very nature directed towards the public because he or she is working with data that belongs to the public.
- lines 54-58: the second part of the sentence seems to be missing a verb
- lines 159-162: splitting this sentence into several distinct phrases would make it clearer to read
Author Response
Comment 1: The plan consists of 4 main parts. Two of them have sub-sections, but I don't understand why there are paragraphs in these parts that don't belong to sub-sections. I think you should rethink this aspect of your paper so as to achieve a better balance between parts and sub-parts.
Response 1: We are sorry, but the authors do not understand the comment. Do you wish for more sub-sections or less paragraphs? Could the reviewer explain which paragraphs are unnecessary or where more subtitles are desired? We used the journals template: maybe, this is a problem for the editors?
For explanation: The 1. Making cultural heritage accessible to young people works as an introduction to the topics being addressed; it is further divided into 1.1. Seven teenagers, lots of ideas! diving into the specific case study. This is followed by 2. Working methods and technical implementation, divided into 2.1. Educational approach and didactic implementation where details on the decisions and approaches are given and 2.2. Through the innermost part of the museum where other aspects of the internship outside of the project-work are displayed. The 3. Output and positive side effects present how the tool is used and its award. 4. Students internship – Is it worth the effort and time? is more related to the initial assessments of the project and conclusions. Although there is not yet sufficient information to further assess the final result presented, the authors believe the mediation tool and the process to create it are positive.
Comment 2: Nevertheless, it is still easy to follow the development of your ideas and they remain very clear. One aspect of your plan that should also improve is the place you give to the museum's presentation. The method refers to the work done on 140-year-old inventories. It is only when the museum is presented in detail in the following sub-sections that we understand why this work is being carried out on these old inventories. The presentation of the museum could therefore come before and be inserted into the sections presenting the context in order to better understand the method used by the students.
Response 2: We agree, please see the new content from 174-183.
Comment 3: A few sentences about the background of the students taking part in the project might help us to understand why they chose to participate and do this internship. For readers unfamiliar with your school system, it is important to specify whether the students are in private or state education. This is essential if we want to analyse the scientific mediation process that has been developed here and help other institutions to draw inspiration from your experience.
Response 3: We agree, please see the new content from 183-204.
Comment 4: A parallel could also be drawn with similar projects.
Response 4: We agree, please see the new content from 441-457.
5: A few details:
- lines 33-34: I fully understand the point being made, but I find it a doubly sharp argument to use this phrase, which is often employed to criticise archaeology and heritage protection institutions. The argument could be turned more towards the fact that archaeology and heritage are the property of everyone, and that the work of the archaeologist is by its very nature directed towards the public because he or she is working with data that belongs to the public.
- lines 54-58: the second part of the sentence seems to be missing a verb
- lines 159-162: splitting this sentence into several distinct phrases would make it clearer to read.
Response 5:
Lines 33-34: We understand the concern and discussed the matter, but we like to keep the sentence sharp. We have considered other formulations the reviewer presented to us, but have concluded that they omit the important detail, that archaeologist do not only work with public data, we get financed by the public. We know about the criticism; the authors also deal with it on the daily basis. However, we believe that a clear address of this fact will show critics that we take the fact seriously and try to make archaeological research and heritage accessible to the public with various measures. We are really trying to live up to the obligation to publish the results and communicate them to the public and would like to implement this idea more strongly in our scientific community.
Lines 54-58 and 159-162: We agree, changes have been made.
Reviewer 3 Report
Comments and Suggestions for Authors
Fonseca and colleagues present an interesting paper on creating digital mediation tools for prehistoric sites. It is a case study, based on the well-known UNESCO World Heritage Site “Prehistoric pile dwellings around the Alps”, a type of prehistoric lakeshore settlement. As the authors indicate, these are difficult to divulge sites, due to their characteristics, thus in need of more uncommon mechanisms for a fruitful and knowledge-rich mediation with the public.
The authors make a good case and introduction to the need to make cultural heritage accessible to young people. A plus is that the presented initiative, not only involved several experts and non-experts, but it was a truly co-creative approach including young interns/students that were the main drivers of this initiative (accompanied by experts). Besides that, the main theme revolves around concepts of recycling and efficient use of resources, hot topics in our society, and to which archaeology should and can give relevant input, as is shown in this paper.
This paper does not deal with a traditional topic nor is it written traditionally. This is not a problem, since it does not hinder the aim of the paper, which is to present a well-conceived example/case study. As such, I cannot review it under the same parameters traditionally used for other manuscripts. Nonetheless, the Abstract is well-prepared. The 1. Making cultural heritage accessible to young people works as an introduction to the topics being addressed (but see below); it is further divided into 1.1. Seven teenagers, lots of ideas! further diving into the specific case study. This is followed by 2. Working methods and technical implementation, divided into 2.1. Educational approach and didactic implementation and 2.2. Through the innermost part of the museum where details on the decisions and approaches are given. The 3. Output and positive side effects and 4. Students internship – Is it worth the effort and time? are more related to the initial assessments of the project and conclusions. Although there is not yet sufficient information to further assess the final result presented, the authors (and I agree) believe the mediation tool and the process to create it are positive.
This paper presents an interesting case study deserving of publication and discussion among peers. Some minor criticism is given to further reinforce the theoretical part of the paper, making it more interesting and accessible to a wider audience, thus more citable:
1. Please give some more information regarding i) the theoretical framework adopted and ii) the archaeological contexts. Concerning i), the paper mostly follows an empirical perspective, but it is clear that a theoretical background guided several of the options made. Can you develop that? As for ii), this is not the main aim of this paper, but some information on these aspects is lacking.
2. Are there no other similar initiatives deserving of mentioning and comparison? Initiatives dealing with citizen science, divulging and communication to different types of the public have been known worldwide for a long time. Although clear parallels are unknown to this reviewer regarding the specific involvement and deliverables here presented, several projects have been developed in the last years trying to produce digital mediating tools, to involve citizens (including students) in different ways in the production of science (not the aim here), but also in the communication of scientific work to participants of initiatives or the public.
3. Please develop a bit more on the selection of the example objects criteria.
4. As a small note, please put the figures after them being mentioned in the text, not before.
I think this paper and the process presented are very interesting, and the interactive digital mediation tool is quite informative in an accessible way. Congratulations to all involved.
Author Response
Comment 1: Please give some more information regarding i) the theoretical framework adopted and ii) the archaeological contexts. Concerning i), the paper mostly follows an empirical perspective, but it is clear that a theoretical background guided several of the options made. Can you develop that? As for ii), this is not the main aim of this paper, but some information on these aspects is lacking.
Response1: We agree, please see the new content from 238-253.
Comment 2: Are there no other similar initiatives deserving of mentioning and comparison? Initiatives dealing with citizen science, divulging and communication to different types of the public have been known worldwide for a long time. Although clear parallels are unknown to this reviewer regarding the specific involvement and deliverables here presented, several projects have been developed in the last years trying to produce digital mediating tools, to involve citizens (including students) in different ways in the production of science (not the aim here), but also in the communication of scientific work to participants of initiatives or the public.
Response 2: We agree, please see the new content from 441-457.
Comment 3: Please develop a bit more on the selection of the example objects criteria.
Response 3: We agree, please see the new content from 238-253.
Comment 4: As a small note, please put the figures after them being mentioned in the text, not before.
Response 4: Thank you, your point is taken and changes have been made.
Comment 5: I think this paper and the process presented are very interesting, and the interactive digital mediation tool is quite informative in an accessible way. Congratulations to all involved.
Response 5: Thank you very much, we really appreciate your comment.